# Antilisterial Properties of Selected Strains from the Autochthonous Microbiota of a Swiss Artisan Soft Smear Cheese

**DOI:** 10.3390/foods13213473

**Published:** 2024-10-30

**Authors:** Alexandra Roetschi, Alexandra Baumeyer, Hélène Berthoud, Lauriane Braillard, Florian Gschwend, Anne Guisolan, John Haldemann, Jörg Hummerjohann, Charlotte Joller, Florian Loosli, Marco Meola, Javorka Naskova, Simone Oberhänsli, Noam Shani, Ueli von Ah, Emmanuelle Arias-Roth

**Affiliations:** 1Fermentation Organisms, Agroscope, 3097 Liebefeld, Switzerlandmarco.meola@databiomix.com (M.M.); noam.shani@agroscope.admin.ch (N.S.); 2Biotechnology, Agroscope, 3097 Liebefeld, Switzerlandueli.vonah@agroscope.admin.ch (U.v.A.); 3Molecular Ecology, Agroscope, 8046 Zürich, Switzerland; florian.gschwend@agroscope.admin.ch; 4Applied Processing Technology, Agroscope, 3097 Liebefeld, Switzerland; 5Microbiological Food Safety, Agroscope, 3097 Liebefeld, Switzerland; 6Cultures, Biodiversity and Terroir, Agroscope, 3097 Liebefeld, Switzerland; 7Interfaculty Bioinformatics Unit, University of Bern, 3012 Bern, Switzerland

**Keywords:** amplicon sequencing, citrate, inhibition, *Ruoffia*, *Marinilactibacillus*, *Desemzia*

## Abstract

High incidences of the foodborne pathogen *Listeria monocytogenes* have been reported on smear cheeses, and despite increased hygiene efforts, this incidence has remained stable in recent years. Applying antilisterial strains may increase the safety of smear cheeses. To find and test antilisterial strains, we inoculated fresh soft cheeses from nine dairies with the surrogate species *Listeria innocua* and assessed its growth under standardized ripening conditions. Acetic acid at day 23 (r = −0.66), lactose in fresh cheese (r = −0.63), and glucose at day 10 (r = −0.62), as well as seven amplicon sequence variants (ASVs), were negatively correlated with *L. innocua* growth. Two of these ASVs were assigned to the genus *Leuconostoc* of Lactobacillaceae (r = −0.82 and −0.71). Isolates from this family, from Aerococcaceae, and Carnobacteriaceae were characterized according to their inhibitory properties, and those showing antilisterial properties were applied as protective cultures in challenge tests. The combined application of strains of *Leuconostoc mesenteroides*, Aerococcaceae, and Carnobacteriaceae successfully eliminated low levels of *L. innocua* in the final products. This is likely explained by antimicrobial compounds, including mesentericin Y105 and acetate, and competition for carbon sources and iron. This study shows a promising way to improve the safety of soft smear cheeses by applying defined protective cultures.

## 1. Introduction

A wide biodiversity of microorganisms colonize natural and artificial habitats [1,2]. Microorganisms are not distributed randomly but are selected by abiotic and biotic factors; each species occupies an ecological niche and exerts a dedicated function in an ecosystem. The role of spontaneous fermentation and back-slopping reflects traditional fermented food practices that rely on naturally occurring microorganisms, fostering unique flavor profiles and textures. Modern hygiene practices can disrupt fermentation ecosystems by removing both harmful and beneficial microbes, making the use of starter cultures essential for restoring desired microbial functions and ensuring consistent quality [3,4,5,6].

Cheese is one of the most popular fermented foods and has numerous varieties. Studies of the cheese microbiota of recent decades using both culture-dependent and -independent methods, such as high-throughput sequencing (HTS) [7,8,9], have been conducted. Beyond starter cultures, autochthonous microbiotas originating from raw materials [10] and production facilities [11,12] have been found in dairy fermented foods, eventually becoming dominant in the final products.

The cheese production process shapes microbiotas and greatly influences their beta diversity [13,14]. In particular, traditional artisan cheeses harbor rich microbiotas [15]. Although the role of autochthonous microbiotas in the cheese-making process remains to be investigated in detail, numerous studies have suggested that autochthonous microbiotas contribute to food safety [16,17,18,19]. They achieve this by inhibiting pathogenic species via several mechanisms, including rapid milk acidification, the production of antimicrobial compounds, ecological competition, or a combination thereof [20]. Inhibition is generally exerted at the microbial community level [21]. For example, the synergy of lactic acid and ripening bacteria was involved in the inhibition of *Listeria monocytogenes* in the core of uncooked pressed cheeses, with higher levels of acetic acid and alcohol measured in inhibitory cheeses [22]. Antilisterial activity has also been linked to specific organisms of cheese smear, for example, Aerococcaceae and Carnobacteriaceae, which grow during early ripening [23], or *Fusobacterium,* which was detected during a product’s shelf life [24].

*L. monocytogenes* is a foodborne human pathogen that affects immunocompromised patients (e.g., the elderly, pregnant women, and newborns) with a very high case fatality rate of 20–30%. An increasing trend of human invasive listeriosis in Europe was reported by the European Food Safety Authority in 2018 (i.e., a 0.7% monthly increase in females over 75 between 2008 and 2015) [25].

Hypervirulent *L. monocytogenes* clonal complex CC1 has been associated with dairy products [26]. Beyond severe cattle illnesses, such as rhombencephalitis and abortions, *L. monocytogenes* is globally persistent in raw milk due to undiagnosed mastitis and fecal carriage by healthy dairy cattle. Its incidence in cheese is linked to the type of cheese and not to milk heat treatments [27], with brined cheeses having the highest incidence (11.8%). Smear cheese also has a high incidence of *L. monocytogenes* (5.1%). Despite joint efforts by farms and dairies to improve hygiene, the incidence of *L. monocytogenes* in smear cheese remains the same as reported in the early 2000s [28]. The level of environmental contamination measured within smear liquid samples was comparable in industrial dairies that had extensive hygienic procedures and in small dairies, with 1.55% and 1.29% positive samples, respectively [29].

Vacherin Mont-d’Or (VMO) is a Swiss artisan soft smear cheese that has been registered with a protected designation of origin (PDO) since 2003. It is produced from thermized milk during the winter season. This moderate heating process enables the control of *Salmonella* spp. but not *L. monocytogenes*, which is more heat-resistant and better adapted to growing in the rind [30]. Indeed, the consumption of VMO was implicated 40 years ago in a listeriosis outbreak [31]. Nowadays, exhaustive control measures for *L. monocytogenes* are implemented in VMO, i.e., the qualitative monitoring of every batch of fully ripened cheeses ahead of their distribution. While hygiene and monitoring are key to guaranteeing pathogen control in food, protective cultures are a promising complementary strategy to improve food safety in cheese types that support the growth of *L. monocytogenes*.

Our study aimed to characterize the antilisterial properties of selected strains from the autochthonous microbiota of VMO, ripened under standardized conditions and intentionally inoculated with a cocktail of *Listeria innocua* strains as surrogates for *L. monocytogenes* [17,32,33]. Therefore, we aimed to (i) quantify *L. innocua* growth under standardized conditions on cheeses from different dairies, (ii) isolate and characterize strains with antilisterial properties, and (iii) validate the antilisterial properties of selected strains on experimental cheeses.

## 2. Materials and Methods

### 2.1. Experimental Design

This study was carried out in three main steps (see Figure 1). In the first step, fresh VMO cheeses from nine dairies were ripened under standardized conditions (Cheese Trial I). Surface-inoculated *L. innocua* was monitored during ripening and storage, concomitant with the development of cheese rind microbiota and biochemical parameters.

In the second step, strains of Aerococcaceae and Carnobacteriaceae isolated from fully ripened VMO, as well as Lactobacillaceae isolated during Cheese Trial 1, were characterized for their antilisterial properties. Therefore, in vitro inhibition tests and whole-genome sequencing were carried out.

In the third step, three factors were investigated in challenge tests carried out on experimental cheeses (i.e., one biochemical factor highlighted in Cheese Trial I as well as two protective cultures made of strains from the VMO autochthonous microbiota and showing in vitro inhibition activity). Two contamination levels of *L. innocua* were considered in the challenge tests: a high level monitored by quantitative analysis during ripening and storage (Cheese Trial II), and a low level monitored by qualitative analysis at the end of storage (Cheese Trial III).

### 2.2. Cheese Trial I

Fresh VMO cheese samples were collected after the brining step in nine Swiss dairies coded with letters A to I throughout this manuscript. In detail, five cheese loaves of 400 to 600 g from a single batch for each dairy were transferred into a sterile plastic bag, stored at 4 °C for two days, transferred to an experimental cellar, and ripened at a relative humidity of 90–92% and a temperature of 12.5 °C (standardized ripening).

The cellar and the brushing procedure were designed to prevent cross-contamination between the dairies (spatial separation of the nine variants A to I, sterilization of the wooden shelves ahead of ripening, regular disinfection of the smear table and apron, change of gloves and brushes for each variant, daily cleaning, and sterilization of the wooden brushes).

The cheeses were turned daily for four days, and subsequently smeared with sterile tap water for the remaining 19 days of the ripening period. Moreover, on days 5–8, samples were smeared with a commercial culture, OMK 702 (Liebefeld Kulturen AG, Liebefeld, Switzerland), consisting of the *Debaryomyces hansenii* yeast and the *Brevibacterium aurantiacum* and *Glutamicibacter arilaitensis* bacteria, dissolved in sterile tap water (1% *v*/*v*), and on days 7–10, with a mixture of 4 *L. innocua* strains (30,000 cfu/mL of sterile tap water), named FAM 20869, FAM 20870, FAM 20871, and FAM 20872 and previously isolated from cheese smears [17]. At the end of the ripening period, the cheeses were individually packed in wooden boxes, which were then put in individual plastic bags and stored at 4 °C.

The whole experiment was replicated three times, i.e., in November 2019 (Dairies A, B, C, D, E, G, H, and I), in February 2020 (Dairies A, C, D, E, F, G, and H), and in February 2021 (all nine dairies).

Analyses were carried out after one (day 1), 10 (day 10), 16 (day 16), and 23 days of ripening (day 23), and at the end of storage period (day 35). The *L. innocua* cell count was monitored on day 10, 16, 23, and 35, in all replications. The biochemical parameters were monitored on days 1, 10, and 23, in all replications. The cheese rind microbiota was investigated on days 10, 16, 23, and 35 in the first and second replications.

### 2.3. Listeria innocua Detection

In Cheese Trials I and II, 10 g of the cheese rind was cut with a sterile scalpel on both flat sides of the cheese loaf. A sample consisted of two slices (each ~20 cm^2^ and a depth ~0.2 cm) pooled together and homogenized in 90 mL 40 °C warm peptone water at a pH of 7.0 (10 g L^−1^ peptone from casein, 5 g L^−1^ sodium chloride, and 20 g L^−1^ tri-sodium citrate dihydrate, all from Merck, Grogg Chemie, Stettlen, Switzerland) for 3 min in a stomacher (Masticator, IUL Instruments GmbH, Troisdorf, Germany). Serial dilutions of the homogenate were then plated on Palcam agar (prepared with selective supplement SR0150, Oxoid, Basingstoke, UK) and ready-to-use ALOA^®^ Agar (bioMérieux, Petit-Lancy, Switzerland) and incubated for 48 h at 37 °C for the *L. innocua* count as described in [23]. Both media were used in parallel to avoid the otherwise necessary biochemical and serological testing of the colonies. In Cheese Trial III, a qualitative enrichment of 25 g of cheese rind was carried out at the end of storage (35 days). In brief, 25 g of rind (2 loaves sampled on both sides; ~100 cm^2^; depth of ~0.2 cm) were pre-enriched in half-Fraser broth, followed by an enrichment in Fraser broth and detection on ALOA^®^ Agar.

### 2.4. Cheese Rind Microbiota Assessment

In Cheese Trial I, DNA was extracted from 1 mL homogenate of the cheese rind, according to [34]. Bacterial microbiota was investigated by performing 16S rRNA gene amplicon sequencing of the V1–V2 region on an Ion PGM™ instrument according to [9]. Taxonomic annotation was carried out using DAIRYdb v3.0.0 [35].

### 2.5. Biochemical Parameters

One hundred g representative samples from the whole cheese, including core and rind, were used to determine biochemical parameters in fresh cheeses. pH was determined using a pH electrode (InLab^®^ Solids Pro-ISM, Mettler Toledo, Greifensee, Switzerland). Water content was determined using a gravimetric method (drying at 102 °C [36]). The concentrations of D- and L-lactate, lactose, galactose, glucose, and citrate were determined using commercial enzymatic assay kits (R-Biopharm AG, Darmstadt, Germany). Those parameters were again measured after 10 days of ripening in Cheese Trial I and after 5 and 10 days of ripening in Cheese Trial III. At the end of ripening in Trial I, free volatile carboxylic acids were determined, as described previously [37].

### 2.6. Isolation of Bacterial Strains

Strains of Aerococcaceae and Carnobacteriaceae were isolated from a smear of fully ripened VMO PDO cheeses produced in eight dairies in 2017 (all but Dairy C). Cheese smear scraped off from 30–40 cm^2^ surface using sterile cotton rolls was homogenized in 30 mL peptone water at room temperature and plated on TGYA agar (Tryptone Glucose Yeast Agar, BD BBL^TM^, supplemented with 1% (*w*/*v*) casein peptone, Merck, Grogg Chemie, Stettlen, Switzerland) or Columbia agar with 5% sheep blood (bioMérieux, Petit-Lancy, Switzerland). In addition to isolation from the smear, 1 mL of brine samples was enriched in 9 mL of AGS-broth8 (Liebefeld Kulturen AG, Liebefeld, Switzerland, confidential information) for 3 days at 30 °C and plated on TGYA agar. A total of 44 isolates from the highest dilutions (above 10^6^ cfu/cm^2^) were identified using 16S sequencing using primers 16SUNI-L and 16SUNI-R, as described previously [38]. Genotypes of 36 strains belonging to the Aerococcaceae and Carnobacteriaceae families were determined using (GTG)_5_-PCR [39]. Only one strain of each genotype was kept per dairy and sample type.

Strains of Lactobacillaceae were isolated from cheese rind samples of Cheese Trial I and plated on MRS agar with lactose instead of glucose (Biolife Italiana, Monza, Italy) or TGYA agar. A total of 110 isolates from the highest dilutions (above 10^6^ cfu/g) were identified using MALDI-TOF [40]. Genotypes of the 89 strains belonging to the Lactobacillaceae families were determined using box C1R-PCR or (GTG)_5_-PCR [39]. One strain of each genotype was kept per dairy and production date.

### 2.7. In Vitro Inhibition Tests

Frozen stocks of all strains were kept in a medium containing sterile low-fat milk as a cryoprotectant. The storage temperature was set at −80 °C for long-term storage or at −40 °C for up to six months.

Seventeen strains of Aerococcaceae and Carnobacteriaceae were propagated twice in AGS-broth8 at 30 °C. Cells of 10 mL overnight cultures (5 × 10^8^–1 × 10^9^ cfu/mL) were concentrated ten times and resuspended in 1 mL of the same media. An overnight culture of the indicator strain *L. innocua* FAM 20870, previously isolated from cheese smear and acting as representative of the cocktail used in cheese trials, was propagated in Brain Heart Infusion Broth (BHI) at 30 °C for 14 h. The indicator strain was inoculated at ~10^6^ cfu/mL into BHI containing 0.7% agar and poured into Petri dishes. Holes 9 mm in diameter were cut into agar media with a sterile pipette tip and filled with 60 μL of resuspended Aerococcaceae and Carnobacteriaceae cells. Plates were placed at 12 °C to enable *L. innocua* growth under conditions mimicking cheese ripening. After four days, the plates were examined for zones of inhibition.

Thirty-eight Lactobacillaceae strains were grown aerobically in MRS broth (De Man, Rogosa Sharpe, Biolife Italiana, Monza, Italy) at 30 °C. The cells of overnight cultures were removed by centrifugation, and crude supernatants were used in an agar well diffusion test based on the assay method described previously [41]. An overnight culture of the indicator strain *L. innocua* FAM 20870 was prepared in TS5, i.e., Tryptic Soy Broth (Becton, Dickinson, and Company, Allschwil, Switzerland) amended with 5% glucose (Merck, Grogg Chemie, Stettlen, Switzerland) and 5% yeast extract (Becton, Dickinson, and Company, Allschwil, Switzerland). The indicator strain was inoculated at ~10^6^ cfu/mL into TS5 containing 1% agar and poured into Petri dishes. Holes 6 mm in diameter were cut into agar media and filled with 50 µL of crude supernatants. Plates were left at 4 °C for two hours to enable the diffusion of the bacteriocins and then placed at 30 °C. The next day, the plates were examined for zones of inhibition.

### 2.8. Whole-Genome Sequencing (WGS) and In Silico Analysis

The WGS of 17 Aerococcaceae and Carnobacteriaceae strains and 23 Lactobacillaceae strains was carried out as described previously [42]. In short, DNA extraction was performed using an EZ1 DNA tissue kit (Qiagen, Hilden, Germany) and a BioRobot EZ1 workstation (Qiagen, Hilden, Germany). Quality control assessments of the extracted DNA, library generation, and sequencing runs were performed on the Next Generation Sequencing Platform, University of Bern, Switzerland. De novo sequencing was performed with “TruSeq DNA PCR-free” or “Nextera Flex” libraries. Libraries were sequenced on an Illumina HiSeq3000 instrument (Illumina, San Diego, CA, USA) or on an Illumina Novaseq 6000 instrument (Illumina, San Diego, CA, USA). Trimmed reads were assembled with SPAdes v.3.14.0 [43], and the assemblies were then uploaded to NCBI with automatic annotation using PGAP. Taxonomic assignment was carried out by calculating average nucleotide identity based on Blast+ ANIb according to [44] as well as by generating a genome blast distance phylogeny tree on the Type Strain Genome Server according to [45]. Genome mining for putative bacteriocin genes was carried out using the online tools antiSMASH [46] and BAGEL4 [47]. Also, the genomes of Lactobacillaceae were checked for the presence of two plasmid-encoded operons, i.e., the mesentericin Y105 operon (Genbank: AY286003.1) using Blastn and the citrate operon (Genbank: AJ132782.2) using Blastx.

### 2.9. Challenge Tests

Challenge tests were carried out on a pilot scale based on results obtained in the previous steps, considering the pH at brining (Factor 1), the addition of a protective culture in the vat milk (Factor 2), and the addition of protective culture in the smearing solution (Factor 3).

#### 2.9.1. Cheese Trial II

This cheese trial aimed (i) to mimic a massive recontamination of the vat milk and the ripening facility; and (ii) to enable quantitative monitoring during ripening and storage. For this purpose, three cheese vats, each filled with 55 L of thermized cow milk (65 °C for 15 s), were transformed per day on two successive production days, yielding six times 20 cheese loaves of 400 g (i.e., 120 loaves). All vats were acidified with a liquid culture of *Streptococcus thermophilus* and *Lactobacillus delbrueckii* subsp. *bulgaricus* (obtained from the Interprofession du VMO AOP). To investigate the impact of Factor 1, 10 loaves per vat were acidified to pH of 5.4, while the 10 remaining loaves were acidified to pH of 5.1, yielding 60 loaves for each pH level. To investigate the impact of Factor 2, two vats per day were inoculated with *L. mesenteroides* FAM 25292, FAM 25293, and FAM 25300 at ~10^3^–3 × 10^4^ cfu/mL, yielding 80 loaves with *L. mesenteroides* and 40 loaves without. The 120 cheese loaves were ripened together in an experimental cellar (spatial separation of 24 ripening batches of five loaves each). The ripening procedures were the same as those described in Trial I. To investigate the impact of Factor 3, a mixture of three strains (i.e., *M. psychrotolerans* FAM 23997, *Ruoffia* sp. FAM 24227, and *Desemzia* sp. FAM 24101) was added to the smearing solution of 12 ripening batches at 10^7^ cfu/mL from the eighth day onwards.

One cheese loaf per ripening batch was analyzed at six time points chosen for investigation, i.e., at arrival in the ripening cellar (day 1); after 7, 10, 15, and 21 days of ripening; and at the end of storage (day 35). *L. innocua* counts were monitored at all time points. The development of protective cultures was monitored by dedicated qPCRs as described in 2.11, after 7 and 10 days for Factor 2 and from the tenth day onwards for Factor 3.

#### 2.9.2. Cheese Trial III

This cheese trial aimed (i) to mimic a realistic scenario of contamination remaining in the vat milk after thermization; and (ii) to assess cheese safety using qualitative monitoring according to current practices.

Slight modifications in the composition of the tested protective cultures were carried out as follows. The protective culture applied as Factor 2 was enriched with four (instead of three) mesentericin Y105-producing strains to include a raw milk isolate showing enhanced in vitro activity. The protective culture applied as Factor 3 was simplified in its composition with two instead of three strains added to the smearing solution, i.e., *M. psychrotolerans* FAM 23997 and *Ruoffia* sp. FAM 24227.

A total of 120 cheese loaves were produced on two successive production days and ripened as 24 separate batches of five loaves, as described in Cheese Trial II. Twelve ripening batches each were investigated for Factor 1 (high versus low pH at brining). Eight ripening batches each were investigated for Factor 2 (protective culture in vat milk at 0, 10^3^, and 10^4^ cfu/mL). Twelve ripening batches each were investigated for Factor 3 (presence or absence of protective culture in smearing solution).

One cheese loaf per ripening batch was analyzed at three time points chosen for investigation, i.e., at its arrival in the ripening cellar (day 1) and after 5 and 10 days of ripening. Thereby, the development of mesentericin Y105-producing strains was monitored by qPCR together with the associated changes in biochemical parameters. Two cheese loaves per ripening batch were necessary for the qualitative monitoring of *L. innocua* in 25 g of rind at the end of storage (day 35).

### 2.10. Cultivation of Bacterial Strains for the Cheese Trials

*L. innocua* strains FAM 20869, FAM 20870, FAM 20871, and FAM 20872 from frozen stocks stored at −80 °C were revitalized in TSY (i.e., Tryptic Soy Broth amended with 5% yeast extract). Each strain was propagated twice in TSY for 16 h at 30 °C, immediately mixed with the other three *L. innocua* strains, and added while fresh to the smearing solution or the vat milk.

Aerococcaceae and Carnobacteriaceae strains were cultured individually in AGS-broth8 (Liebefeld Kulturen AG, Liebefeld, Switzerland, confidential information) at 30 °C. FAM 23997 and FAM 24101 were incubated for 16 h, while FAM 24227 was incubated for 24 h. Fresh Aerococcaceae and Carnobacteriaceae cultures were enumerated on M17 agar (Biolife Italiana, Monza, Italy) supplemented with 5 g/L glucose, mixed to the appropriate ratio, and stored for up to seven days at 4 °C before inoculation of the smearing solution. In Cheese Trial II, FAM 23997, FAM 24227, and FAM 24101 strains were inoculated at 3 × 10^6^ cfu/mL each. In Cheese Trial III, FAM 23997 and FAM 24227 were inoculated at 5 × 10^6^ cfu/mL each.

In Cheese Trial II, FAM 25292, FAM 25293, and FAM 25300 were incubated individually in EM-Lmc-45 (Liebefeld Kulturen AG, Liebefeld, Switzerland, confidential information) flasks for 16 h at 30 °C, immediately mixed, and inoculated in vat milk at ~1 × 10^3^–3 × 10^4^ cfu/mL. The cultivation procedure was adapted for Cheese Trial III to precisely monitor the inoculation concentration. Therefore, FAM 24179, FAM 25292, FAM 25293, and FAM 25300 were cocultured in EM-Lmc-45 for 12 h at 25 °C and pH of 5.9. The biomass was concentrated 10 times and frozen as pellets. Pellets were kept at −20 °C for up to two months until inoculation in the vat milk at 1 × 10^3^ or 1 × 10^4^ cfu/mL, respectively.

### 2.11. Real-Time PCR Quantification (qPCR) in the Cheese Matrix

DNA was extracted from a 1 mL homogenate of the cheese rind or 1 mL homogenate of the cheese smear as described above. To monitor the growth of the protective cultures in the cheese rind, primers and probes were designed for the detection of (1) *Marinilactibacillus psychrotolerans*, (2) *Desemzia* sp., (3) a subset of strains from *Ruoffia* sp. (see Appendix A), and (4) the mesentericin Y105 precursor, as described in Appendix A. Quantification was carried out by qPCR using plasmid-based standard curves, as detailed in Appendix A [48].

### 2.12. Statistical Analysis

Two statistical tests were used to assess the growth of *L. innocua* on cheeses from different dairies. First, the difference in *L. innocua* counts during ripening and storage was tested using a linear mixed-effects model implemented in lme (package nlme v3.1-162, [49]) based on the mean values of two or three replications, including “Dairy” as a random-effect variable. Second, the *L. innocua* growth on cheeses from different dairies was tested by comparing the differences in *L. innocua* counts after 10 and 35 days among the dairies. For this, the same function was used, and “replication” was included as a random effect variable to account for the dependence of samples from the same replications.

The development of ASV richness during ripening and storage was tested in the same way for the two replications, including “Dairy” as a random effect variable. Correlations between the two factors were also calculated in R version 4.2.3, using Pearson correlations. *p*-values were adjusted using the Benjamini–Hochberg correction implemented in R. To compare the number of ASVs (i.e., the ASV richness among the samples), we standardized the sequence numbers in each sample by random subsampling to the minimum sequence number in a sample. ASV richness was calculated as the mean of 1000 subsamplings. Microbial community comparisons were calculated based on Bray–Curtis dissimilarities, and differences among replications, dairies, and ripening time points were tested using permutational multivariate analysis of variance (PERMANOVA) implemented in PRIMER7 [50]. To account for repeated measurements over time, a nested PERMANOVA design was chosen following the example given in the PRIMER7 PERMANOVA+ manual [50].

## 3. Results

### 3.1. Cheese Trial I: Factors That Influence L. innocua Growth in the Rind of VMO Cheese

#### 3.1.1. The Rind of VMO Cheese Supports the Growth of *L. innocua* During Ripening in a Facility-Dependent Manner

*L. innocua* counts significantly increased during cheese ripening (*p* < 0.0001), and no significant increase was observed during storage (*p* = 0.95, Figure 2). The growth of *L. innocua*, measured as the difference in *L. innocua* counts between days 10 and 35, varied significantly among the cheeses delivered by the different dairies (*p* = 0.005). While the strongest increase in *L. innocua* was observed on the cheese delivered by dairy F, no significant growth was observed on the cheese delivered by dairy D.

#### 3.1.2. *L. innocua* Counts Correlate with Multiple Biochemical Parameters

Sixteen biochemical parameters were measured during ripening in Cheese Trial I (Appendix A). Correlations between *L. innocua* counts at 35 days (the end of the storage period) and the biochemical parameters at different stages of the cheese-making process were calculated. Negative correlations were found for acetic acid in ripened cheese after 23 days (−0.66; Table 1), residual lactose in fresh cheeses (r = −0.63), and glucose at 10 days (−0.62). Positive correlations were found for lactate (0.62) and L-lactate (0.61) in fresh cheese and for citrate at 10 days (0.55). Two out of sixteen biochemical parameters were significantly different between dairies (*p* < 0.02). These were (1) acetic acid in ripened cheese, which was higher in Dairy D than in Dairy F; and (2) dry loss in fresh cheese, which was higher in Dairy C than in Dairies E and G.

#### 3.1.3. Development of a Rich Autochthonous Microbiota During Ripening in an Experimental Cellar, Including Seven ASVs Negatively Correlated with *L. innocua* Counts

Amplicon sequencing targeting the V1–V2 region of the 16S rRNA gene yielded a total of 15,767,834 quality filtered sequences, which corresponded to 567 amplicon sequence variants (ASVs). These ASVs were assigned to four phyla, eight classes, 18 orders, 36 families, 71 genera, and 103 species. A single species included an average of 3.3 ASVs, but the maximum number of ASVs assigned to a single species was 23 for *Lactobacillus delbrueckii*. Thirty ASVs were found in all cheese rinds, regardless of the dairy delivering the fresh cheeses. These core ASVs were classified into 18 species and included the four species used as starter or secondary cultures in VMO production (Appendix A).

The relative abundance of ASVs assigned to the starter taxa decreased on average and relative to their abundance in 10-day-old cheese by 52.4% (*Lactobacillus delbrueckii*) and 82.2% (*Streptococcus thermophilus*) during ripening and storage (Appendix A). Bacterial communities were composed of 33 to 144 ASVs in a sample, and the bacterial ASV richness increased during the ripening phase by 66.5% on average (Figure 3a,c). The community structure of cheese rinds also changed significantly during cheese ripening and storage (PERMANOVA, *p* = 0.001, Table 2; Figure 3b,d). Bacterial community structures differed markedly between the two replications (PERMANOVA, *p* = 0.001, Table 2), with only 27.0% of all ASVs (153 ASVs) detected in both replications. However, these corresponded to 85.4% of the relative abundance, revealing the consistent presence of the dominant bacterial taxa.

The repeatedly detected ASVs (i.e., those occurring in both replications) were screened for potential *L. innocua* antagonists by correlating *L. innocua* counts in the cheese samples after storage with the relative abundances of ASVs. This yielded seven ASVs, which were significantly and negatively correlated with *L. innocua* counts, with correlation values ranging from −0.82 to −0.71 (Table 1, Appendix A). These seven ASVs were assigned to the following taxa, which were in order of correlation strength *Leuconostoc carnosum/mesenteroides* with two ASVs (r = −0.82 and −0.71), *Weissella hellenica* (r = −0.79), *Latilactobacillus curvatus* with two ASVs (r = −0.77 and −0.74), *Marinomonas* sp./*flavescens*/*ushuaiensis* (r = −0.73), and *Celerinatantimonas* (r = −0.73).

### 3.2. Isolation and Characterization of Putative Antagonistic Strains

#### 3.2.1. Aerococcaceae and Carnobacteriaceae Show Strain-Specific Antilisterial Properties

The smears from fully ripened VMO PDO cheeses collected in eight dairies were investigated for the presence of Aerococcaceae and Carnobacteriaceae (Table 3). The species *M. psychrotolerans* was the most frequently isolated Carnobacteriaceae species, with isolates for six out of the eight dairies. *Desemzia* sp., an undescribed species related to *Desemzia incerta* (ANIb value ≤ 77.4; Appendix A), was the second most frequently isolated Carnobacteriaceae species, with isolates for four out of the eight dairies. Finally, one strain of the Aerococcaceae species, *Ruoffia* sp., an undescribed species related to *Ruoffia* (formerly *Facklamia*) *tabacinasalis* and *Ruoffia halotolerans* (Appendix A), could be isolated from the smear of Dairy E.

Quantification by qPCR showed that the three species were all present in the smear of the eight dairies at various levels (median value 2 × 10^8^, 4 × 10^7^, and 3 × 10^8^ copies per cm^2^ for *M. psychrotolerans*, *Desemzia* sp., and a subset of *Ruoffia* sp., respectively; Appendix A). Cheese brines from Dairies G and I showed a high amount of *M. psychrotolerans*, with >1 × 10^6^ copies per ml of brine (Appendix A). Isolates were obtained from the operating cheese brines in both dairies (Table 3). Smear and brine isolates collected 9 months apart were closely related with an ANIb value > 99.9 (Dairy G: FAM 23998 and FAM24231; Dairy I: FAM 24106 and FAM24230).

All tested Carnobacteriaceae strains showed an in vitro inhibition of *L. innocua*. The inhibition intensity was strain-specific (Table 3) but was not associated with the presence of specific bacteriocin genes. Indeed, the best-performing *Desemzia* sp. strain had no detectable bacteriocin genes in its genome. Similarly, the best-performing *M. psychrotolerans* strains generally had no gene cluster coding for a known bacteriocin in their genomes.

The only Aerococcaceae strain isolated from VMO (i.e., *Ruoffia* sp. FAM 24227) showed a high in vitro antilisterial effect (Table 3). A single gene cluster coding for putative bacteriocin genes (lanthipeptide class IV, Table 3) was present in its genome.

The three best-performing strains, i.e., *M. psychrotolerans* FAM 23997, *Desemzia* sp. FAM 24101, and *Ruoffia* sp. FAM 24227, were selected for challenge tests on the pilot scale. Finally, the results of Biolog phenotype microarrays showed the utilization of five carbon sources potentially present on cheese surfaces —pyruvic acid, D-tagatose, N-acetyl-D-glucosamine, D-trehalose, and D-mannose—by *L. monocytogenes* and at least one of the three antagonistic strains (Appendix A).

#### 3.2.2. Lactobacillaceae Antilisterial Properties

The rind from cheeses collected in Cheese Trial I was investigated for the presence of Lactobacillaceae (Table 4). Nine species were colonizing the rind of cheeses at the dominant level, i.e., in order of isolation frequency: *Latilactobacillus curvatus* and *Leuconostoc mesenteroides*, *Leuconostoc carnosum*, *Latilactobacillus sakei* and *Weissella hellenica*, *Limosilactobacillus fermentum*, *Lacticaseibacillus paracasei*, *Lactiplantibacillus plantarum*, and *Loigolactobacillus zhaoyuanensis*.

Among them, only one of the nine species (i.e., *L. mesenteroides*) demonstrated antilisterial activity in a strain-specific manner (Table 4). An inhibition zone > 2 mm was visible for all strains harboring the plasmid-encoded mesentericin Y105 operon. In particular, the four *L. mesenteroides* strains isolated from cheeses produced in Dairy D carried this plasmid, including two isolates sharing >99.9% ANIb values and originating from VMO batches produced in two successive winter seasons. In the genome of raw milk isolate FAM 24179 producing a wider inhibition zone of >6 mm, an additional gene cluster coding for a class II bacteriocin was present (Table 4).

Finally, the best-performing strains, FAM 24179, FAM 25292, FAM 25293, and FAM 25300, were selected for challenge tests on the pilot scale. Three of these four strains harbored the plasmid-encoded citrate utilization operon (Table 4).

### 3.3. Challenge Tests at the Pilot Scale: The Effect of pH at Brining and the Addition of Protective Cultures in the Vat Milk/Smearing Solution

#### 3.3.1. Worst-Case Scenario: Cheese Trials II: Conditions Mimicking a Massive *Listeria* Recontamination of the Vat Milk and the Ripening Facility

The impact on *L. innocua* growth of three factors was investigated on experimental cheeses: (1) pH level at brining, (2) the addition of mesentericin Y105-positive *L. mesenteroides* strains in the vat milk, and (3) the addition of Aerococcaceae and Carnobacteriaceae strains in the smearing solution. A high pH at brining first favored the growth of *L. innocua* after 10 days. However, this led to lower *L. innocua* counts in the final product, both at the end of ripening and at the end of storage (Figure 4a, Appendix A).

Mesentericin Y105-positive *L. mesenteroides* added to the vat milk grew within 7 days and led to a lower *L. innocua* count during ripening (Figure 4b,d). The protective effect was significant on days 15 and 21. Aerococcaceae and Carnobacteriaceae added to the smearing solution grew during ripening and storage (Figure 4e–g). Their protective effect was significant during storage, as *L. innocua* was unable to grow in their presence (Figure 4c).

#### 3.3.2. Realistic Scenario: Cheese Trial III—Conditions Mimicking a Low *Listeria* Level Remaining in Vat Milk After the Thermization of Contaminated Raw Milk

The pH at brining had no impact on the presence of *L. innocua* in the final products, with 42% of the loaves positive for *L. innocua* at a pH of 5.4 and 50% at a pH of 5.1 (*p* = 1.0; Appendix A).

The addition of mesentericin Y105-positive *L. mesenteroides* strains in the vat milk had a significant impact (*p* < 0.002, Fisher’s exact test with a simulated *p*-value) on the presence of *L. innocua* at the end of storage, with 100% positive samples within the control loaves and 25% positive samples within the loaves treated with a low *L. mesenteroides* dosage and 12.5% positive samples within loaves treated with a high *L. mesenteroides* dosage (Figure 5a). The growth of mesentericin Y105-positive *L. mesenteroides* in the rind was assessed by qPCR (Figure 5b). Its growth after 10 days depended on the pH at brining (two-sample *t*-test, *p* = 0.007) and not on the dosage in the vat milk. A higher pH at brining seems to have supported additional *L. mesenteroides* growth, possibly through the co-metabolism of residual glucose and citrate into D-lactate (Figure 5c,d).

The addition of Aerococcaceae and Carnobacteriaceae strains in the smearing solution only tended to lower the percentage of positive loaves, with 58% positive in the control and 33% positive in the treated loaves (*p* = 0.4136, Figure 5a). However, it is noticeable that the eight loaves with the combination of the protective cultures in the vat milk (i.e., *L. mesenteroides*) and the smearing solution (i.e., Aerococcaceae, and Carnobacteriaceae) were all free of *L. innocua* at the end of the storage period, while the four control loaves without protective cultures were all positive (*p* = 0.002, Fisher’s exact test with a simulated *p*-value).

## 4. Discussion

Bacterial succession through ripening and storage was characterized for the first time in the rind of cheeses produced in VMO dairies. As the cheeses were ripened under standardized conditions and their surfaces were inoculated with *L. innocua*, the bacterial development may be different from that taking place in VMO PDO ripening facilities, which harbor a wider range of abiotic and biotic factors (i.e., temperature, relative humidity, and additional autochthonous microbiotas). The core microbiota of the VMO ripened under standardized conditions consisted of 18 species detected by amplicon sequencing. As already observed in other surface-ripened cheeses [51], the applied starter cultures (*Streptococcus* and *Lactobacillus* spp.) were gradually replaced by secondary cultures (*Glutamicibacter* and *Brevibacterium* spp.) as well as by a rich biodiversity of autochthonous bacteria. The bacterial communities of VMO ripened under standardized conditions share several similarities with Mont-d’Or PDO (MO) [14], a French soft smear cheese. Indeed, their core microbiomes have 12 genera in common, of which nine species are autochthonous to VMO.

In particular, the *Latilactobacillus* genus is the first and second most abundant genus in the rind of MO and VMO, respectively. The two cheese types also belong to the genus *Desemzia*, which was found to be specific to MO when compared to 11 French artisan soft cheese varieties using the same HTS pipeline [14]. It is interesting to note that a rich and partly common flora is established in these two cheese varieties sharing a similar recipe, despite the distinct heat treatment of the milk, i.e., thermized for VMO and raw for MO. Microbial transfers from raw milk or processing surfaces to artisan cheese have been revealed in numerous cheese microbiota studies ([52]; for a review, see [15]). In our study, the source of autochthonous microbiota was not systematically investigated, but some information could be acquired for two bacterial species. Cheese brine was the likely source of *M. psychrotolerans* in two VMO dairies, as smear isolates and operating brine isolates sampled 9 months apart showed an ANIb value of >99.9. Similarly, very closely related *L. mesenteroides* strains were isolated from the rind of cheeses produced in Dairy D in two successive winter seasons, suggesting a long-term colonization of the milking or processing environment.

Putative antilisterial properties were detected by amplicon sequencing within the autochthonous microbiota of VMO. In particular, all ASVs assigned to the genus *Leuconostoc* were negatively correlated with *L. innocua* counts. *Leuconostoc* spp. have been described in complex antilisterial consortia growing in the core and the rind of soft cheeses [15,53]. To our knowledge, our study is the first to suggest a substantial contribution of this genus to the observed antagonistic properties in situ. Low-abundant ASVs assigned to *Weissella hellenica* and *Latilactobacillus curvatus* were negatively correlated with *L. innocua* counts. However, other more abundant ASVs assigned to those two species did not follow the same trend, suggesting that these species played a secondary role, if any, in the observed antagonistic properties. Finally, two ASVs assigned to the Gammaproteobacteria genera *Marinomonas* and *Celerinatantimonas* showed a negative correlation with *L. innocua* counts. Strains of *Marinomonas* have been shown to contribute to the antilisterial properties exerted by model communities isolated from Livarot cheese [54,55].

The in vitro characterization of antilisterial properties was carried out for isolates of Lactobacillaceae, Aerococcaceae, and Carnobacteriaceae. Strain-specific activities were detected within the VMO core species *L. mesenteroides*, *Ruoffia* sp., *Desemzia* sp., and *M. psychrotolerans*. Genome mining revealed that in vitro active *L. mesenteroides* strains all harbored the complete plasmid pFR38, which is involved in the production of the antilisterial Class IIa bacteriocin mesentericin Y105 [56].

For the other three genera, no clear association was detected between clusters coding for putative bacteriocins and in vitro inhibition activities. As their antilisterial activity was later confirmed in situ, the utilization of carbon sources by the three strains selected for the challenge tests was investigated using Biolog phenotype microarrays. A comparison with previously published *L. monocytogenes* data [57] highlighted five carbon sources present on cheese surfaces—pyruvic acid, D-tagatose, N-acetyl-D-glucosamine, D-trehalose, and D-mannose—that may contribute to the antilisterial properties of the Aerococcaceae and Carnobacteriaceae strains in situ through ecological competition for carbon sources. This aspect has yet to be investigated and deserves focused research efforts. Recent studies have instead emphasized essential minerals, such as iron, as limiting factors that shape the microbiota of cheese rinds [58,59].

The role of two biochemical parameters as in situ hurdles against *L. innocua* was highlighted in our study: citrate consumption and acetate production by the autochthonous microbiota of VMO. Similar findings have been described for a complex antilisterial consortium from the rind of St. Nectaire cheese [53]. The antimicrobial activity of organic acids, such as lactate and acetate, has been well documented [20,60]. The role played by citrate is less clear but could be linked to one or more of the following mechanisms. First, a high citrate level could facilitate iron uptake by *L. monocytogenes* using the citrate-induced system [61]. Second, citrate can be converted to diacetyl, which acts as an antimicrobial compound above 100 ppm [62]. Third, the co-metabolism of citrate and glucose could lead to a lower glucose amount in cheese, thereby preventing its use by *L. monocytogenes*. The co-metabolism of citrate and glucose has a positive influence on *L. mesenteroides* growth, as described in a genome-scale metabolic network [63]. In our study, we observed enhanced growth via the co-metabolism of citrate and glucose, with the concomitant production of D-lactate by *L. mesenteroides*.

This current study revealed that autochthonous bacteria reduce the colonization properties of *L. innocua* in the rind of VMO. In particular, the core species *Ruoffia* sp., *Desemzia* sp., and *M. psychrotolerans* generally prevented the growth of *L. innocua* during storage at refrigeration temperature. The antilisterial activity of another mixture of Aerococcaceae and Carnobacteriaceae, i.e., including the genus *Alkalibacterium* instead of *Desemzia*, has been shown to reduce *L. innocua* growth in Raclette-type cheese [23]. Not all dairies benefit equally from the antilisterial properties of autochthonous bacteria. Indeed, only one out of the nine dairies consistently hosted antilisterial Lactobacillaceae. An adjustment of pH levels at brining can be used by dairies as a lever to boost the colonization of rinds by *L. mesenteroides*.

Another approach could be to add mesentericin Y105-producing *L. mesenteroides* to vat milk in low amounts (i.e., mimicking the natural inoculation of raw milk). Strains of the genus *Leuconostoc* have long been applied as protective cultures in meat products (*L. carnosum* [64,65]). To our knowledge, no protective culture with *Leuconostoc* spp. has been specifically developed for dairy products.

Finally, our study showed how the antilisterial autochthonous bacterial species can act in a sequential manner (i.e., Lactobacillaceae at mid-ripening and Aerococcaceae and Carnobacteriaceae during cold storage), leading to the complete elimination of *L. innocua* in the final product.

## 5. Conclusions

In this study, we characterized the antilisterial properties of selected strains from the autochthonous microbiota of VMO. The quantitative monitoring of *L. innocua* in the cheese rind showed that VMO fresh cheeses from one of the nine dairies did not support *L. innocua* growth during ripening and storage under standardized conditions. The genus *Leuconostoc* was detected as a potential *L. innocua* antagonist. The in vitro screening of isolates from the VMO microbiota revealed strain-specific antilisterial properties within the species *Leuconostoc mesenteroides* and the previously described potential antagonist families Aerococcaceae and Carnobacteriaceae. The most effective strains were subsequently tested as protective cultures in situ. The protective cultures demonstrated activity at specific stages of ripening and storage with Lactobacillaceae being most active at mid-ripening, while Aerococcaceae and Carnobacteriaceae had their peak activity during cold storage. Only their combined application resulted in the successful elimination of low levels of *L. innocua* in the final products. Taken together, our results suggest three complementary strategies to increase or introduce antilisterial properties in the VMO rind microbiota: (i) an adjustment to a higher pH at brining to boost the colonization of the rind with *L. mesenteroides*, (ii) the addition of mesentericin Y105-producing *L. mesenteroides* to the vat milk in low amounts, and (iii) the application of selected strains of Aerococcaceae and Carnobacteriaceae to counteract *Listeria* growth during storage.

## Figures and Tables

**Figure 1 foods-13-03473-f001:**
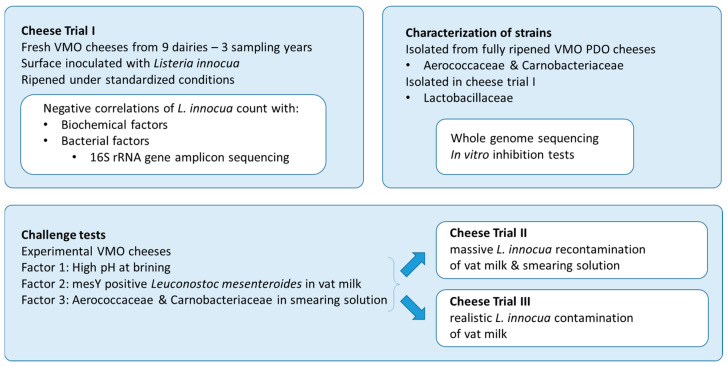
Experimental design.

**Figure 2 foods-13-03473-f002:**
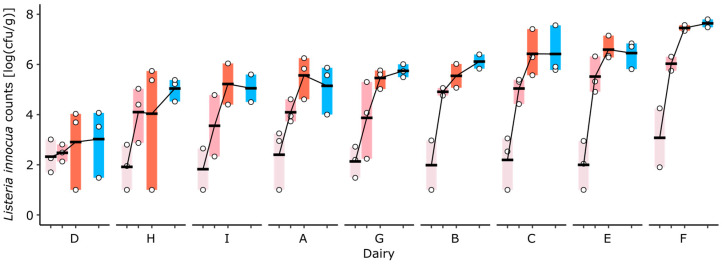
The growth of *L. innocua* in the rind of VMO cheeses produced in nine dairies (A–I) and ripened in an experimental cellar under standardized conditions. Colors indicate the sampling time points during ripening at days 10 (light rose), 16 (rose), and 23 (red), as well as after cold storage at day 35 (blue).

**Figure 3 foods-13-03473-f003:**
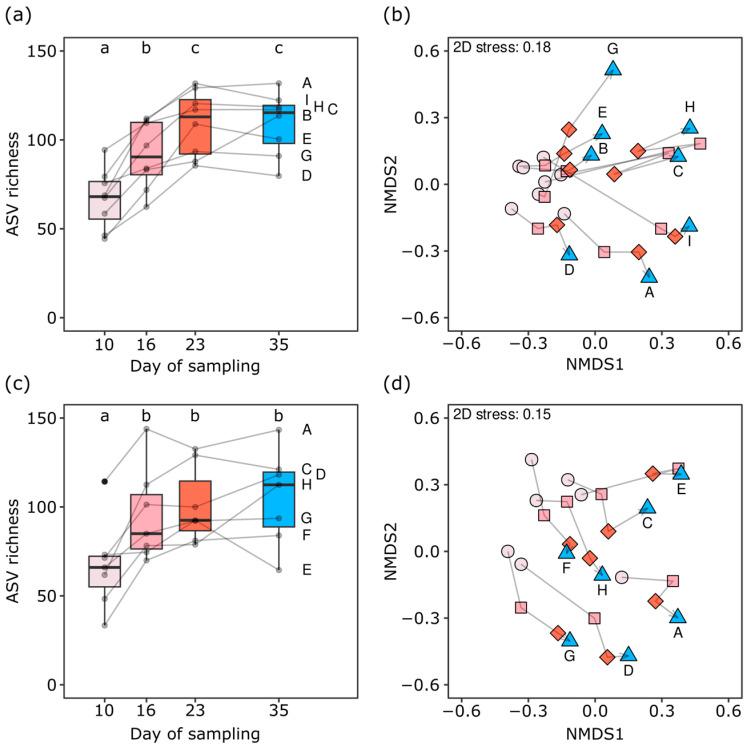
Diversity of bacterial communities developed on rinds of VMO cheese ripened in an experimental cellar. ASV richness (**a**,**c**) and community structures (**b**,**d**) are shown. The trial was repeated twice, with panels (**a**,**b**) showing the results of the first replication and panels (**c**,**d**) showing the second replication. Colors and shapes of points in (**b**,**d**) indicate the sampling time points during ripening at days 10 (light rose, circle), 16 (rose, square), and 23 (red, diamond) as well as after two weeks of cold storage at day 35 (blue, triangle). Gray lines connect the measurements obtained for cheeses produced in the same VMO dairies, which are also labeled with capital letters. Lowercase letters above the boxplots indicate significance groups (adj. *p* < 0.05). The ordination of the community structures was calculated by nonmetric multidimensional scaling based on Bray–Curtis dissimilarities among communities.

**Figure 4 foods-13-03473-f004:**
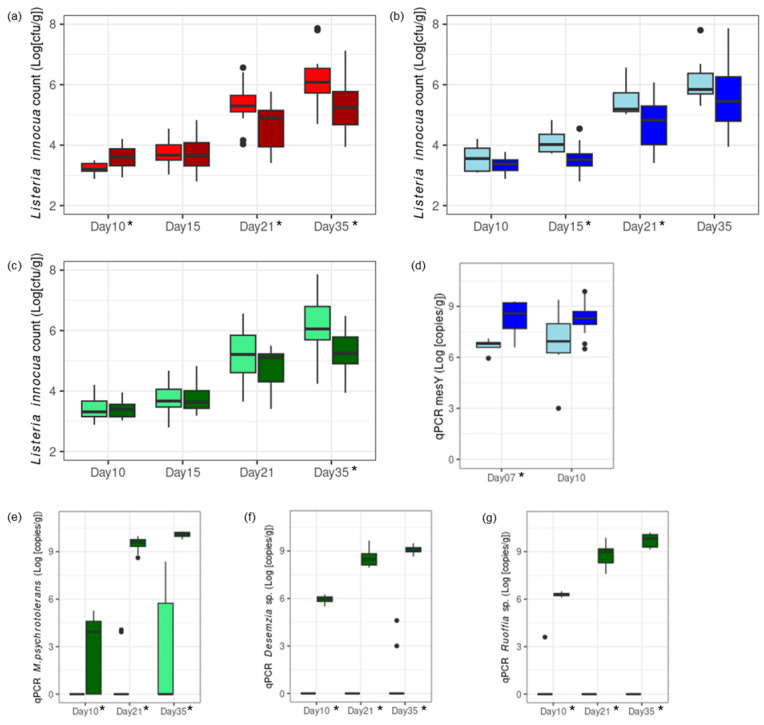
Cheese Trial II. Challenge tests at pilot scale with a high degree of *L. innocua* contamination were conducted for the investigation of three factors: (1) low pH at brining (pH 5.1, red; panel (**a**)) versus a high pH at brining (pH 5.4, dark red; panel (**a**)); (2) no addition of protective culture in the vat milk (light blue; panels (**b**,**d**)) versus addition of mesentericin Y105-positive (mesY) *L. mesenteroides* in the vat milk (blue; panels (**b**,**d**)); and (3) no addition of protective culture in the smearing solution (light green; panels (**c**,**e**–**g**)) versus addition of Aerococcaceae and Carnobacteriaceae in the smearing solution (dark green; panels (**c**,**e**–**g**)). The growth of *L. innocua* (cell counts; panels (**a**–**c**)) and protective cultures (qPCR; panels (**d**,**e**–**g**)) were measured at relevant time points along ripening (days 7–21) and storage at 4 °C (day 35). Time points at which significant differences were measured (two-sample *t*-test, *p* < 0.05) are indicated with stars.

**Figure 5 foods-13-03473-f005:**
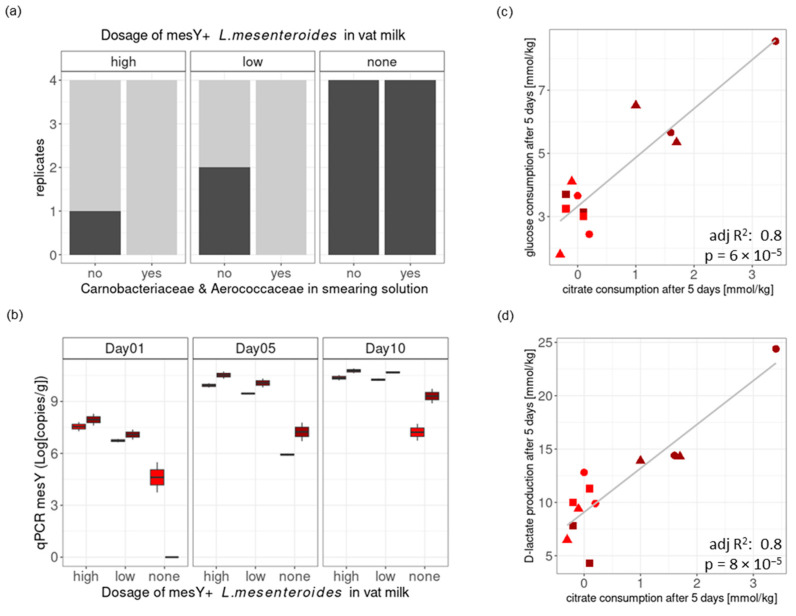
Cheese Trial III. Challenge tests at pilot scale with low degree of *L. innocua* contamination of the vat milk. Panel (**a**): The presence (dark gray) or absence (light gray) of *L. innocua* was assessed in 25 g of cheese rind at the end of storage (35 days). Mesentericin Y105-positive (mesY) *L. mesenteroides* strains were tested at three dosages: high, low, or none. The addition of Aerococcaceae and Carnobacteriaceae strains to the smearing solution was also tested. Panel (**b**): The level of mesY-positive *L. mesenteroides* in the smear was assessed by qPCR at three time points (1, 5, and 10 days). Positive correlations were observed between citrate consumption and glucose consumption (panel (**c**)) as well as between citrate consumption and D-lactate production (panel (**d**)), as measured between fresh cheese and cheese after five days of ripening. The colors of points in (**b**–**d**) indicate the pH at brining (red, low pH; dark red, high pH). Shapes of points in (**c**,**d**) indicate the dosage of mesentericin Y105-positive *L. mesenteroides* (high, circle; low, triangle; none: square).

**Table 1 foods-13-03473-t001:** Correlations of *L. innocua* counts at the end of storage (35 days) with biochemical factors and bacterial ASVs. All detected significant correlations are shown (adj. *p* < 0.05). Stars indicate significance levels: (*) *p* < 0.05, (**) *p* < 0.01.

Factor	r	*p*-Value	adj. *p*-Value	
Biochemical				
Acetic acid at 23 days	−0.66	0.000	0.013	*
Lactose in fresh cheese	−0.63	0.001	0.013	*
Glucose at 10 days	−0.62	0.001	0.013	*
Citrate at 10 days	0.55	0.005	0.029	*
L-lactate in fresh cheese	0.61	0.002	0.013	*
Lactate in fresh cheese	0.62	0.001	0.013	*
Bacterial				
ASV_005 *Leuconostoc carnosum/mesenteroides*	−0.82	0.000	0.004	**
ASV_212 *Weissella hellenica*	−0.79	0.001	0.010	*
ASV_147 *Latilactobacillus curvatus*	−0.77	0.001	0.017	*
ASV_110 *Latilactobacillus curvatus*	−0.74	0.002	0.031	*
ASV_068 *Marinomonas* sp./*flavescens*/*ushuaiensis*	−0.73	0.002	0.032	*
ASV_227 *Celerinatantimonas*	−0.73	0.002	0.032	*
ASV_007 *Leuconostoc carnosum/mesenteroides*	−0.71	0.003	0.047	*

**Table 2 foods-13-03473-t002:** Differences among bacterial community structures of cheese rinds on cheeses produced by nine different VMO dairies used for Cheese Trial I. Two replications of the cheese trial were performed, and three time points were sampled during cheese ripening (i.e., on days 10, 16, and 23) as well as after cold storage on day 35.

	Df	R^2^	Pseudo-F	*p*-Value	√CV ^1^
Dairy	8	39.6%	1.1	0.3703	0.09
Replication	6	27.5%	9.9	0.0001	0.39
Cheese ripening and at storage	3	13.4%	9.7	0.0001	0.20
Residuals	42	19.4%			0.26
Total	59	100%			

^1^ CV: component of variation, i.e., the difference among factor levels expressed in Bray–Curtis dissimilarity.

**Table 3 foods-13-03473-t003:** Aerococcaceae and Carnobacteriaceae strains are sorted by genus, source, and dairy. Phenotype and putative bacteriocin genes are indicated.

Strain ^1^	Source ^2^	Dairy	Inhibition of *L. innocua* by Live Bacteria ^3^	Putative Bacteriocin Genes ^4^
Aerococcaceae				
*Ruoffia* sp.				
FAM 24227 *	VMO	E	+++	B1
Carnobacteriaceae				
*Desemzia* sp.				
FAM 23990	VMO	B	+	B2
FAM 23991	VMO	B	+	no
FAM 24101 *	VMO	D	+++	no
FAM 23989	VMO	F	++	no
FAM 23988	VMO	H	+	B3
*Marinilactibacillus psychrotolerans*				
FAM 24229	VMO	A	++	no
FAM 23992	VMO	D	(+)	no
FAM 23995	VMO	F	(+)	B4
FAM 23631	VMO	G	+	B4, B5, B6
FAM 23997 *	VMO	G	++	B7
FAM 23998	VMO	G	(+)	B8
FAM 23993	VMO	H	+	no
FAM 24102	VMO	H	+	B7
FAM 24106	VMO	I	++	no
FAM 24231	Cheese brine	G		B8
FAM 24230	Cheese brine	I		no

^1^ Strains selected for cheese trials based on their in vitro antagonistic phenotype are marked with a star (*). ^2^ Isolates were sampled from smears or brines in Vacherin Mont-d’Or PDO (VMO) dairies. ^3^ Inhibition zone was classified as (+) >0 and ≤2 mm, + >2 and ≤4 mm, ++ >4 and ≤6 mm, +++ >6 mm. ^4^ Closest bacteriocin types detected by genome mining were B1: lanthipeptide class IV, B2: Linocin M18, B3: Subtilosin A, B4: Listeriolysin S, B5: lanthipeptide class II, B6: Enterocin W, B7: Sakacin P, B8: putative bacteriocin.

**Table 4 foods-13-03473-t004:** Lactobacillaceae strains isolated in Cheese Trial I sorted by genus and dairy that delivered fresh VMO cheese. Phenotypes and genes of interest are indicated.

Strain ^1^	Dairy	Inhibition of *L. innocua* by Supernatant ^3^	Mesentericin Y105 Operon ^4^	Other Putative Bacteriocin ^5^	Citrate ^6^ Operon
*Lacticaseibacillus paracasei*					
FAM 25336	D	(−)			
*Lactiplantibacillus plantarum*					
FAM 25263	D	(−)	0	B9	incomplete
*Latilactobacillus curvatus*					
FAM 25311	A	(−)			
FAM 24637	D	(−)	0	no	no
FAM 25309	D	(−)			
FAM 25316	E	(−)			
FAM 25313	G	(−)			
FAM 25314	H	(−)	0	B10	yes (>52%)
FAM 25315	H	(−)			
*Latilactobacillus sakei*					
FAM 24915	E	(−)	0	B11	yes (>53%)
FAM 24914	G	(−)	0	B11	yes (>53%)
FAM 24916	I	(−)	0	no	incomplete
*Leuconostoc carnosum*					
FAM 24634	D	(−)	0	B12	yes (>67%)
FAM 24918	D	(−)			
FAM 25319	D	(−)			
FAM 25322	G	(−)			
FAM 25323	G	(−)			
FAM 24917	H	(−)			
FAM 24919	I	(−)			
*Leuconostoc mesenteroides*					
FAM 25299	A	(−)	0	B13	no
FAM 25300 *	A	+	100	B13	no
FAM 24636	D	+	100	B13	yes (>99%)
FAM 25292 *	D	+	100	B13	yes (>99%)
FAM 25293 *	D	++	100	B13	yes (>99%)
FAM 25295	D	+	100	B13	no
FAM 25301	G	(−)	0	B13	no
FAM 25302	G	(−)	0	B13	no
FAM 25281	H	(−)	0	B13	no
FAM 25285	I	(+)	0	B13	no
FAM 24179 *^,2^	-	+++	100	B14	yes (>98%)
*Limosilactobacillus fermentum*					
FAM 25337	D	(−)	0	B15, B16	incomplete
FAM 25338	H	(−)	0	B15, B16	no
*Loigolactobacillus zhaoyuanensis*					
FAM 25317	E	(−)	0	B17	incomplete
*Weissella hellenica*					
FAM 25328	D	(−)			
FAM 25329	F	(−)			
FAM 25330	F	(−)	0	no	no
FAM 25332	G	(−)	0	no	no
FAM 25333	G	(−)			

^1^ Strains selected for cheese trials based on their in vitro antagonistic phenotype are marked with a star (*). ^2^ This isolate was sampled from raw milk collected in Switzerland. ^3^ Inhibition zone was classified as (−) 0 mm, (+) >0 and ≤2 mm, + >2 and ≤4 mm, ++ >4 and ≤6 mm, +++ >6 mm. ^4^ The percentages of identical base pairs are indicated. ^5^ Closest bacteriocin types detected by genome mining were B9: Plantaricin E and F; B10: Curvacin A; B11: putative bacteriocin; B12: putative bacteriocin; B13: Enterocin Xbeta; B14: class II bacteriocin; B15 and B16: Enterolysin A; B17: Sactipeptides. ^6^ Percentage of positive amino acids for citM, citC, citD, citE, citF, citG, citR, and citP are indicated; no: operon not detected; incomplete: citrate permease citP is missing.

## Data Availability

The amplicon sequencing dataset generated and analyzed during this current study is available in the Sequence Read Archive under BioProject number PRJNA1148047. The WGS dataset is available in the GenBank genetic sequence database under BioProject number PRJNA1149749.

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
