# Peer review of "Antilisterial Properties of Selected Strains from the Autochthonous Microbiota of a Swiss Artisan Soft Smear Cheese"

_foods, 2024, doi:10.3390/foods13213473_

Round 1
Reviewer 1 Report
Comments and Suggestions for Authors
The article is poorly structured, they talk about a type of cheese with a designation of origin but do not say which one. The introduction is not clear, and it is also a type of cheese about which there is a lot of information. The materials and methods could be specified using a flow chart to make them more intuitive, Table 1 shows too much information. The results are not clear, the figures are confusing and the discussion of the results is too poor. The results must be supported by other articles for which there is too much information. Furthermore, they do not provide conclusions or bibliographic references, so the article cannot be accepted for publication.
Comments on the Quality of English Language
Moderate editing of English language required.
Reviewer 2 Report
Comments and Suggestions for Authors
Dear authors, I read carefully your manuscript about “Autochthonous microbiota contributes to the safety of a traditional Swiss red smear cheese”. The subject of the manuscript is of interest due to the great importance given the L. monocytogenes impact on human health, mainly because if effectively inactivated by pasteurization, L. monocytogenes postprocessing contamination of ready-to-eat food and dairy products has been well established and its control in processing plant environments is thus critical.
However, I found several deficiencies in the study design and the methodological approach, which might affect the possibility of publication in the Foods journal, unless a drastic revision of the manuscript is carried out.
My main concern is the study design and the sample size used. Indeed, no information is provided on the cheese studied, the sample units, or the control samples eventually used as reference. The methods section is very confusing, with a lack of detail that prevents the reader from interpreting the various tests performed. Indeed, the different reported trials lack specifications (Trials IIa and IIb what are they referring to?)
The second critical point concerns the authors’ reference to the indigenous microbiota of a rind cheese that has been inoculated. In my opinion, it is not possible to refer to the rind's autochthonous microbiota when the rind cheese has previously been inoculated with an L. innocua strains mixture, because the structure of the natural microbial population could be modified. Unless the rind cheese microbiota study as well as the strain isolation have not been carried out, on the positive controls. But this detail is not revealable in the text, as it should be. Moreover, the introduction is unfocused, with more emphasis on the biodiversity of the ecosystems than is warranted given the scope of the study, without any reference to the type of cheese being studied, to its production process, or to the microbial groups that usually characterize it or its safety. Moreover, no mention of L. innocua, as a surrogate for L. monocytogenes, was made (Why did not inoculate L. monocytogenes?). In many cases, no reference support of their statements throughout the manuscript was used. Lastly, although explanatory, the last part of the introduction is too long.
Under the specific line comments, I did not distinguish between major and minor problems, because moderate English editing of the overall text and considerable modification in the structure are required. Furthermore, where possible, it is necessary to modify and align the caption format and the figures with the text.
Other specific comments are reported below.
Title - the study aimed to characterize the antilisterial properties of a group of selected strains isolated from an experimental Swiss artisan soft smear cheese produced with thermised milk and inoculated with L. innocua cocktail strains. Modify the title according to this comment.
Abstract - this section is vague. The authors should be rewritten again because it does not represent the paper sufficiently, taking also into consideration the aforementioned comments.
Line 86 - the authors should introduce a paragraph on the experimental design to overall explain the study steps and purposes.
Line 88 – in this paragraph, the authors should provide details on L. innocua mixture strains, what is meant by replication (perhaps several tests were carried out to have more samples available?), and why dairies change during the course of replication. Moreover, the authors should provide details on the sample size determination and sampling periods (hours and days from?). How many cheese samples were produced and analyzed?
Line 109 – why the end of storage is 35 and not 38? (23 ripening days + 15 days of storage).
Line 111: the authors should explain the nature of Trial IIa, IIb, and III, never before mentioned. They should justify the utilization of two kinds of media for listeria cultivation (comprised of the mention of production specifications), specify the peptone water mL used, and the details of the companies where the media were purchased.
Lines 124 and 131 - the authors should explain the selection of only some sampling periods for the microbiota analyses of the cheese rind as well as for the biochemical tests.
Line 138 - the author should specify the grams used for the analyses of the cheese smear as well as the protocols of MALDI TOF, GTG5, and BOXC1R, perhaps simply with bibliographic references. Moreover, the here introduction of the two tables, containing the information isolates, is misleading because results are introduced even before all analyses have been specified. Instead of tables that can be inserted before the challenge test paragraph or in the results section, it would be appropriate to specify the overall isolates number and of these how many were more in-depth characterized. Why strains isolated in other tests were included in Tables 1 and 2?
Line 174 – in this paragraph, the authors should specify more details on the L. innocua strain (origin, type strain) and the Aerococcaceae and Carnobacteriaceae cell density, as well as clarify the choice of two different kinds of medium to cultivate the same strain (L. innocua), and the different inoculum volume.
Line 195 – the authors should report the number of selected strains subjected to WGS and the reason why different techniques were used.
Line 215 – in this paragraph, the authors should explain the selection criteria used for choosing the microorganisms to be inoculated into the various cheese trials as well as their inoculum size.
Line 242 - the authors should explain the reasons for the two specific pHs selected.
Lines 239 and 271 - the authors should better specify the selection criteria of the two challenge test scenarios used.
Lines 256 and 267 – the authors should explain the reasons for changes in inoculation and ripening procedures.
Results
Compared to the previous sections, the results are better structured, but in some cases redundant because they take up parts that should have been specified in the materials and methods section. In general, it would be advisable to limit this section to results description and try to follow the order of paragraphs as in the previous section.
Line 322 – the authors should be reviewing results, because L. innocua did not always remain constant during cheese storage.
Figure 1 - the description of the experiment in the figure caption is superfluous. The authors should limit themselves to explaining what is shown in the figure.
Line 354 – the authors should better explain from where and how many strains were subjected to WGS as well as refer them to the rind cheese microbiota.
Discussion
The discussion section is better structured and well integrates their findings. However, as in the introduction, there are again places where the references provided are lacking and do not accurately represent the statement being made. Authors should extend the discussion section by giving or comparing the results of the present and previous studies.
Line 547 – the authors should explain to what microbiota refer because were carried out analysis for the rind cheese microbial population as well as for strains isolated from both smear cheese and brine and report speculation following this data.
Line 548 - the authors should explain why the detection of Ruoffia sp. was possible by qPCR rather than sequencing.
Conclusion
The conclusion section should be revised to give the right prominence to the comprehensive study carried out, highlighting only the importance of the obtained results.
Comments on the Quality of English Language
A moderate overall revision of the English in the overall text would be appropriate
Round 2
Reviewer 1 Report
Comments and Suggestions for Authors
The authors have made the requested corrections and the article can be considered for publication
Comments on the Quality of English Language
Moderate editing of English language required.
Author Response
Comments and Suggestions for Authors
The authors have made the requested corrections and the article can be considered for publication
Dear reviewer,
we would like to thank you for your second review of the manuscript. The line numbering indicated below refers to the pdf file, in which major changes are indicated in red.
The abstract has been entirely rewritten to follow a more defined structure and introduce results from the biochemical tests (lines 22 to 37).
The microbial density labelling was adapted by converting the qPCR results given in copies per reaction into copies/mL (liquids), log copies/g (cheese rind) and copies/cm2 (cheese smear).
Finally, the conclusion has been extended to highlight the promising strategies to increase or introduce antilisterial properties in the VMO rind microbiota (lines 678 to 683)
Reviewer 2 Report
Comments and Suggestions for Authors
Dear authors,
thank you for following all my suggestions. The introduction of the experimental design has certainly improved the possibility of understanding by the readers, but I find that some sections of the work need further improvement. I have inserted suggestions and requests directly in the text.

Comments on the Quality of English Language
Minor editing of the English language is required
Author Response
Comments and Suggestions for Authors
Dear authors,
thank you for following all my suggestions. The introduction of the experimental design has certainly improved the possibility of understanding by the readers, but I find that some sections of the work need further improvement. I have inserted suggestions and requests directly in the text.
Dear reviewer, we would like to thank you for the additional suggestions. We have made changes throughout the manuscript in response to your comments. The line numbering indicated below refers to the pdf file, in which major changes are indicated in red.
The abstract has been entirely rewritten to follow the indicated structure (lines 22 to 37). Moreover, results obtained from the biochemical tests are now integrated (lines 27-28).
Two paragraphs of the introduction were rewritten (line 44-49) or replaced (96-99) as suggested.
The sentences of the material and methods section were simplified, reorganised and partly renamed as suggested (paragraphs 2.2, 2.7, 2.9).
The sentences of the results section were rewritten as suggested (lines 354-359, lines 467-470, lines 481-482 and lines 502-503).
We kept the first paragraph of the discussion because we think that it is important to state that our ripening conditions does not reflects the variability in the PDO ripening facilities.
The conclusion has been extended to highlight the promising strategies to increase or introduce antilisterial properties in the VMO rind microbiota (lines 678 to 683)
Answer to the comment : “standardize microbial density labelling in all manuscript (number of cells or log CFU/g or mL).”
We agree with the reviewer that microbial density labelling can be better standardized and we adapted the labelling with the following structure:
- All dosage in liquids (medium, milk, brine, smearing solution) are given in cfu/mL or copies/mL.
- In one case, we adapted the density labelling from “copies per reaction” into “copies per mL” (line 443). To account for the dilution steps and volume transformation during sample preparation, values in copies per reaction were multiplied by 1 to gives copies per mL of brine.
- All bacterial levels in the rind are now given in log CFU/g or log copies/g.
- In the case of the qPCR results, we adapted the density labelling “log copies per reaction” into “log copies per g” (Figure 4 and 5; Tables S7 and S8). To account for the dilution steps and volume transformation during sample preparation, values in copies per reaction were multiplied by 500 to gives copies per g of rind.
- The bacterial levels in the smear are given in copies/cm2.
- We adapted the density labelling “copies per reaction” into “copies per cm2” (lines 440, Table S6). To account for the dilution steps and volume transformation during sample preparation, values in copies per reaction were multiplied by 50 to gives copies per cm2 of smear. As the smear sampling was carried out based on a surface and not a weight, it is not possible to give a density in copies per g.